# The Data Management Crisis Behind AI-Based Brain MRI Diagnosis: Heterogeneity, Governance, and Reproducibility

Tanmoy Debnath
Charles Sturt University
Bathurst, New South Wales, Australia
tdebnath@csu.edu.au

Miroslaw Narbutt
Technological University Dublin
Dublin, Ireland
miroslaw.narbutt@TUDublin.ie

Katarzyna Anna Kazimierczak
University of Bergen
Bergen, Norway
katarzyna.a.kazimierczak@uib.no

## ABSTRACT

AI-based brain magnetic resonance imaging (MRI) classifiers face a persistent gap between laboratory performance and clinical deployability. We argue the bottleneck is not model architecture but the absence of principled data management infrastructure. Drawing on our ongoing work in explainable multi-disease MRI classification, we identify five critical data management challenges: heterogeneity, governance misalignment, pipeline irreproducibility, annotation provenance, and demographic bias. To address these issues, we propose a minimal proof-of-concept metadata harmonization toolkit as a practical and tractable starting point for standardization. We further outline a validation protocol that measures field-level mapping coverage, clinician-reviewed preservation of phenotype and severity-score semantics, provenance completeness, and downstream classifier transfer across ADNI and OASIS-3. We invite the data management community to co-design the schema-mapping and query-optimization layers.

**VLDB Workshop Reference Format:**
Tanmoy Debnath, Miroslaw Narbutt, and Katarzyna Anna Kazimierczak. The Data Management Crisis Behind AI-Based Brain MRI Diagnosis: Heterogeneity, Governance, and Reproducibility. VLDB 2026 Workshop: Biomedical Data Management Systems (BioDMS).

## 1 INTRODUCTION

Brain diseases—encompassing neurodegenerative disorders (Alzheimer's, Parkinson's, amyotrophic lateral sclerosis (ALS) and related motor neuron diseases) and primary brain tumors—represent a leading cause of disability-adjusted life years globally. MRI is a central non-invasive modality for brain assessment, yet manual interpretation is time-consuming and subject to substantial inter-observer variability for early-stage presentations. Deep learning approaches, particularly residual convolutional architectures and transfer learning pipelines, have demonstrated strong benchmark performance [4, 8, 24]. However, the gap between laboratory results and clinical deployability has persisted for over a decade [31]. Our ongoing project developing explainable, multi-disease MRI classifiers reveals that the principal bottleneck is not model architecture but the absence of robust supporting data-management

infrastructure. Recent advances in continuous integration for machine learning (ML), data-quality verification, healthcare analytics, cohort discovery, and medical-imaging privacy converge on a common conclusion: reliable clinical AI requires robust infrastructure for data validation, integration, querying, governance, and reproducible execution. This body of work collectively demonstrates that such systems, rather than increasingly complex predictors alone, are essential for trustworthy deployment in clinical settings [5, 6, 18, 19, 21, 32, 34]. We present five problem clusters as concrete open problems for the data systems community.

## 2 FIVE DATA MANAGEMENT CRISES

### 2.1 Multi-Source Heterogeneity

Major neuroimaging repositories—including Alzheimer's Disease Neuroimaging Initiative (ADNI) [17], the Open Access Series of Imaging Studies (OASIS) [22], IXI, and the Brain Tumor Segmentation benchmark (BraTS) [24]—differ substantially in acquisition protocols (field strength, echo time, repetition time, voxel resolution), scanner manufacturer, and patient demographics. Table 1 summarizes five widely used repositories based on public documentation for ADNI, OASIS-3, BraTS, IXI, and UK Biobank.

Figure 1 illustrates how clinical severity metadata is represented differently across ADNI, OASIS-3, and BraTS, highlighting incompatibilities in naming, coding, instrument choice, and recording granularity.

Metadata schemas in clinical imaging remain inconsistent and often incomplete, with acquisition parameters, scanner calibration records, and clinical phenotype definitions variably encoded or missing altogether. Existing ontology efforts, e.g., NIDM and RadLex, address only a narrow subset of the problem and fall well short of the breadth, maturity, and adoption of the Gene Ontology in genomics, while longitudinal phenotyping and severity-score support remain sparse and non-mandatory. Federated querying, i.e., asking one query across data that remain distributed across institutions, is therefore infeasible without bespoke, manually curated harmonization scripts that neither generalize across studies nor are version-controlled. Prior data-systems work on semantic typing, table embeddings, table-union search, and biomedical identifier mapping demonstrates the infrastructure needed for robust cross-repository discovery [7, 16, 20, 32].

### 2.2 Governance Misaligned with AI Training

Access conditions remain fragmented: repositories impose differing data use agreements, institutional registration requirements, and embargo periods. Cross-border training exposes unresolved tensions between the General Data Protection Regulation (GDPR)'s

**Table 1: Comparison of five major neuroimaging repositories. Label granularity: scan (S), subject (Su), and cohort (C).**

| Repository | Access model | BIDS | Labels | Preproc. documented |
|---|---|---|---|---|
| ADNI | Restricted | Partial | Su | Partial |
| OASIS-3 | Restricted | Partial | Su | Partial |
| BraTS | Open | No | S | None |
| IXI | Open | No | None | None |
| UK Biobank | Tiered | No | Su | Partial |

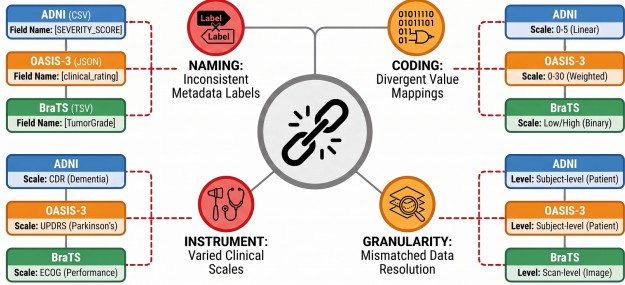

**Figure 1: Schema divergence in clinical severity metadata across ADNI, OASIS-3, and BraTS**

strict purpose-limitation regime and the Health Insurance Portability and Accountability Act (HIPAA)'s comparatively broader secondary-use allowances, while divergent transfer mechanisms create governance friction that cannot be resolved through administration alone. The resulting gap calls for technical controls that can enforce policy, provenance, and data-use constraints across jurisdictions. Federated benchmarking, biomedical data-discovery interfaces, and privacy-preserving medical-image analysis suggest partial solutions but reveal that governance, cohort selection, and secure computation must be designed together [1, 2, 6, 21, 23].

## 2.3 Preprocessing Pipeline Irreproducibility

Neuroimaging pipelines (registration, intensity normalization, atlas-based segmentation) are sensitive to software versions (FSL, FreeSurfer, ANTs, etc.) and parameter configurations. The choice of skull-stripping tool alone, e.g., FSL BET, HD-BET, or ANTs, produces measurably different brain masks that propagate divergent volumetric estimates downstream. These pipelines are rarely containerized or publicly deposited, making independent replication practically impossible: performance metrics cannot be unambiguously attributed to architecture versus preprocessing choices versus dataset composition. Continuous integration for ML, automated data-quality verification, validity constraints, and hybrid provenance systems provide concrete building blocks [3, 9, 18, 19, 30].

## 2.4 Annotation Provenance and Ground Truth

Supervised multi-disease classification requires clinician-verified, scan-level labels. Existing datasets either lack such labels, aggregate multi-rater annotations without recording inter-rater reliability or adjudication protocol, or provide no machine-readable link to the

scoring instruments from which labels derive. Severity scoring requires ordinal labels referenced to validated clinical instruments (e.g., Alberta Stroke Program Early CT Score (ASPECTS) for stroke, Clinical Dementia Rating (CDR) for dementia, and Unified Parkinson's Disease Rating Scale (UPDRS) for Parkinson's disease), yet this linkage is rarely maintained in queryable, structured form. For severity and phenotype variables, correctness is not merely syntactic field alignment; mappings should preserve the clinical instrument, scoring range, score directionality, assessment date, rater or adjudication status, and diagnostic context.

Biomedical mapping resources and cohort-discovery systems show that portable and auditable clinical labels require explicit identifier mappings, curation workflows, and interpretable cohort definitions [5, 16]. Medical-image labeling workflows also need data models that record label creation, reviewer decisions, annotator disagreement, and label reliability [25, 29, 33].

## 2.5 Ethical Risks from Demographic Bias

Public MRI repositories are not demographically representative: ADNI recruited predominantly non-Hispanic white, college-educated participants [17], and similar skews are documented across major repositories. This systematic under-representation risks AI systems that perpetuate healthcare disparities. Addressing this is a data governance problem—requiring curation-level demographic auditing, dataset documentation, and bias-aware quality checks—as much as an algorithmic one. Recent data-quality work explicitly connects error detection and debugging with fairness and robustness, supporting the view that demographic bias should be monitored as part of the data lifecycle rather than treated as a post-hoc model issue [19].

## 3 REMAINING GAPS

Brain Imaging Data Structure (BIDS) provides a widely adopted convention for organizing and describing MRI datasets, including structural MRI, resting-state fMRI, and task-fMRI, but clinical disease metadata (severity scores, comorbidities, medication history) remains sparsely covered and non-mandatory. Moreover, repositories such as ADNI and OASIS-3 are often distributed in DICOM or repository-specific formats, while BIDS-compatible versions typically depend on additional conversion and curation workflows. OpenNeuro [27] offers BIDS-validated open sharing but is oriented toward cognitive neuroscience, with no mechanism to link scans to electronic health records. DataLad [14] addresses provenance tracking and version control for large datasets, but it does not resolve clinical metadata harmonization, governance alignment, or cross-repository query semantics. fMRIPrep [12] and sMRIPrep [11] advance preprocessing standardization for functional and structural MRI, respectively, while MRIQC [10] provides automated image-quality assessment; however, these tools do not resolve clinical metadata harmonization, governance alignment, annotation provenance, or cross-repository query semantics.

Federated learning frameworks, e.g., FeTS [28] demonstrate that cross-site training is technically feasible but presuppose harmonized site data—precisely the unsolved problem. Data-systems research offers important component solutions: robust ingestion [32], semantic typing, table embeddings [7], table search [20], biomedical

**Table 2: Infrastructure coverage gaps for clinical neuroimaging AI. Coverage is based on public documentation; "Target" denotes planned proof-of-concept functionality.**

| INFRASTRUCTURE | Clinical Metadata | Provenance & Versioning | Cross-repository Query | Governance & Access | EHR Linkage |
|---|---|---|---|---|---|
| BIDS | Sparse | Partial | No | No | No |
| OpenNeuro | Partial | No | No | Open Only | No |
| DataLad | No | Yes | No | No | No |
| fMRIPrep/sMRIPrep | No | Yes | No | No | No |
| MRIQC | No | Partial | No | No | No |
| FeTS | No | No | No | Yes | No |
| Proposed Toolkit | Target | Target | Target | Target | Target |

semantic mappings [16], data-quality verification, workflow validity constraints [30], provenance systems [3], cohort discovery [5], disease-centric portals [1], and scalable biomedical indices.

Table 2 positions the proposed toolkit against existing infrastructure and shows that no current system jointly supports the metadata, provenance, query, governance, and linkage capabilities required for multi-disease neuroimaging AI.

## 4 PROOF-OF-CONCEPT DIRECTION

The challenges above are tractable open research problems in data integration, metadata management, provenance tracking, and federated query optimization—areas of core VLDB and SIGMOD expertise. Five immediate targets are:

(1) A unified, extensible neuroimaging metadata schema compatible with Fast Healthcare Interoperability Resources (FHIR), Digital Imaging and Communications in Medicine (DICOM), and Brain Imaging Data Structure (BIDS), with mandatory clinical phenotype fields and machine-readable severity-score linkage supported by biomedical semantic mapping resources [13, 15, 16, 26].

(2) Query languages and optimization strategies for heterogeneous, partially open, multi-site repositories with differing access tiers and schema conventions [5, 7, 20, 32].

(3) Provenance-aware pipeline management enforcing containerization, data-quality verification, validity constraints, and reproducible re-execution across sites [3, 18, 19, 30].

(4) Data models for annotation uncertainty, inter-rater disagreement, adjudication history, and identifier-level clinical label provenance in medical image labeling workflows [29, 33].

(5) Privacy-preserving federated learning infrastructure aligned with biomedical ethics governance, removing the need for pre-harmonized site data [1, 6, 21, 23].

**Minimal proof-of-concept direction.** We propose a lightweight community prototype—a *metadata harmonization toolkit*—that (i) ingests repository-specific manifests (e.g., ADNI CSV, OASIS JSON, and BraTS TSV files), (ii) maps fields to a common BIDS-extension schema using embedding-based ontology alignment and

LLM-assisted field disambiguation, and (iii) generates a versioned, queryable SQLite index with provenance links to the source files.

Figure 2 presents the proposed metadata harmonization toolkit, from repository-specific manifest ingestion to ontology-aware field alignment, provenance capture, and construction of a versioned queryable index over immutable source records.

The prototype requires no central storage or duplication of raw imaging data. It will be evaluated on a small cross-repository Alzheimer's disease cohort constructed from ADNI and OASIS-3. To ensure a falsifiable assessment, evaluation will be conducted at three levels. First, metadata harmonization performance will be quantified by reporting the numbers of source fields ingested, automatically mapped fields, manually validated fields, ambiguous fields, and fields rejected as semantically unsafe. Second, clinical-semantic validity will be assessed by verifying diagnostic labels, severity measures, phenotype definitions, and temporal variables against the original repository data dictionaries and, where feasible, through review by a clinician or neuroimaging domain expert. Third, downstream utility will be evaluated by comparing cohort construction and machine learning model transfer before and after harmonization, reporting cohort overlap, missingness reduction, balanced accuracy, AUROC, calibration, and cross-repository generalization. Figure 3 summarizes the planned validation framework, linking harmonization quality to clinical-semantic validity and downstream utility in cohort construction and classifier transfer.

Table 3 summarizes the proposed evaluation framework and the corresponding metrics used to assess the utility, validity, and auditability of the harmonization toolkit. We invite data management researchers and repository maintainers to contribute to the co-design of the schema-mapping and query-optimization components and to participate in the community-driven evaluation of the framework.

**Loss-aware harmonization.** Beyond maintaining provenance links, a key design principle of the proposed toolkit is that harmonization should not overwrite or replace source metadata. Instead, harmonized metadata are represented as a queryable view over immutable source records. Each canonical field retains links to the source field, original value, source instrument, unit of measurement, transformation rule, mapping confidence score, reviewer

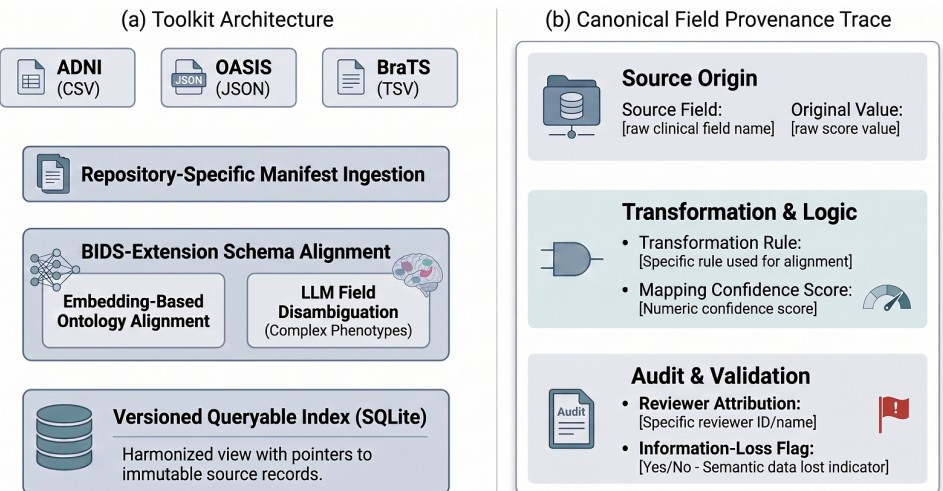

**Figure 2: (a) Proposed architecture of the metadata harmonization toolkit. (b) Provenance trace for one canonical field, preserving the source field, original value, transformation rule, mapping confidence, reviewer attribution, and information-loss flag.**

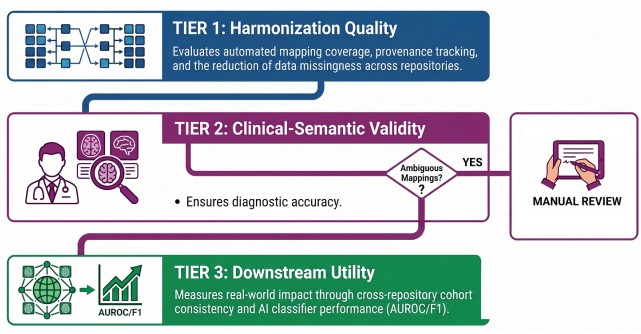

**Figure 3: Planned three-tier validation framework.**

**Table 3: Evaluation dimensions for the toolkit.**

| Target | Metric | Goal |
|---|---|---|
| Schema mapping | % fields mapped automatically/manually/rejected | Demonstrates toolkit coverage beyond a conceptual proposal. |
| Clinical meaning | Clinician agreement on severity and phenotype mappings | Tests preservation of the source instrument's meaning. |
| Provenance | % fields with source and transformation trace | Ensures auditability and reproducibility. |
| Data quality | Missingness before versus after harmonization | Shows practical value for cohort construction. |
| Query utility | Cross-repository cohort-query consistency | Demonstrates cross-repository discovery. |
| Classifier utility | AUROC, F1, balanced accuracy before versus after harmonization | Connects infrastructure to diagnostic utility. |

attribution, review status, and an explicit information-loss flag. Ambiguous mappings remain accessible for inspection and querying but are excluded from clinical-label evaluation until manually reviewed. This loss-aware design makes semantic uncertainty and information loss explicit rather than implicit, thereby improving transparency, auditability, and reproducibility. Furthermore, every harmonized value can be traced directly to its repository-specific origin, enabling downstream researchers to verify transformations and assess the impact of harmonization decisions.

## 5 PROPOSED TALK AND COLLABORATION

This lightning talk will present concrete examples from harmonizing ADNI, OASIS-3, and BraTS for multi-disease classification. It will include preliminary mapping coverage and information-loss tracking outputs, where available, and will conclude with problem statements designed to initiate dialogue on co-developing infrastructure solutions. We seek to establish a working group to develop a metadata harmonization prototype and a community benchmark. This benchmark will consist of a curated cross-repository cohort with harmonized metadata, documented preprocessing provenance, clinical-semantic validation of labels and severity scores, and downstream classifier-transfer checks, released under permissive licenses. Algorithmic progress alone cannot resolve these structural impediments; collaboration with the data management community is essential.

## ACKNOWLEDGMENTS

The authors express their gratitude to the anonymous reviewers for their insightful comments and rigorous critique, which improved the quality of this manuscript. Furthermore, we extend our appreciation to the workshop organizers and the Program Committee for their efforts in facilitating the event.

## AUTHORS

**Dr. Tanmoy Debnath** is a Senior Lecturer in Engineering at Charles Sturt University, Australia. His research interests include artificial intelligence, digital health, medical imaging, and data science. From 2024 to 2026, he secured competitive funding for four research projects in AI, e-health, and data science, acting as the principal investigator on two projects. His research has been published in Q1 journals, CRC Press (USA) book chapters, and workshops affiliated with leading CORE A* conferences, including ICDM 2025, SIGMOD 2026, KDD 2026, and VLDB 2026.

**Dr. Miroslaw Narbutt** is a lecturer at Technological University Dublin, Ireland. He holds a PhD in Computer Science from University College Dublin, Ireland, and an MSc in Electronics and Telecommunications Engineering from Poznan University of Technology, Poland. His research interests include real-time multimedia transmission (WebRTC), immersive audio (Ambisonics), deep learning methods in medical diagnostics, and generative audio. Miroslaw Narbutt is the inventor of two patented technologies, one of which is used by YouTube/Google.

**Dr. Katarzyna Kazimierczak** received her PhD in Neuroscience from the University of Bergen, Norway, where she is currently a postdoctoral researcher. She specialises in MRI data analysis, including functional (fMRI), structural (sMRI), and magnetic resonance spectroscopy (MRS) using MATLAB, Python, and specialised brain imaging software, with research focused on advancing neuroimaging methods for clinical and cognitive neuroscience.

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
