# OpenReview forum: "The Data Management Crisis Behind AI-Based Brain MRI Diagnosis: Heterogeneity, Governance, and Reproducibility"
_VLDB.org/2026/Workshop/BioDMS — BioDMS 2026 LightningTalk_

### Official Review · Reviewer_HCKk · 2026-06-01

**Summary:**

The authors identify five critical data management challenges for explainable multi-disease MRI classification, paired with a "call-to-action" and a proposal for concrete collaboration directions.

**Confidence Of Review:**

3

**Detailed Feedback Points:**

S1. The paper uncovers several deep technical problems in research data management for MRI classification.

S2. The paper proposes concrete research directions that are actionable on a short-term.

I don't see any major opportunities for improvement, I think this a great example of the kind of paper that this workshop should attract.

**Relevance For Biodms:**

4

---

### Official Review · Reviewer_2HM2 · 2026-06-04

**Summary:**

The paper presents five data management challenges that hinder the deployment of AI-based MRI classifiers in clinical settings. These challenges are heterogeneity, governance misalignment, pipeline irreproducibility, annotation provenance, and demographic bias. The authors present directions for solving these problems based on the systems and approaches from data management.

**Confidence Of Review:**

3

**Detailed Feedback Points:**

S1: Great proposal for the proof-of-concept, all based on existing technologies from data management. There is clearly a huge potential of collaboration.

W1: Missing reference in Section 3.

W2: Here and there, the language is too domain-specific, but doesn't impede the understanding.

**Relevance For Biodms:**

4

---

### Official Review · Reviewer_Rjfr · 2026-06-17

**Summary:**

This paper argues that AI tools for brain MRI diagnosis rarely reach the clinic not because the models are inadequate but because the data infrastructure around them is heterogeneous, poorly governed, and irreproducible, and it sets out five specific problems: inconsistent metadata across repositories, misaligned governance and data-use rules, poorly documented preprocessing pipelines, weak provenance, and demographic bias in the underlying cohorts. To address these, the authors propose a lightweight metadata harmonization toolkit that maps repository-specific fields onto a common schema and outputs a versioned, queryable index, framed as an invitation for the data management community to collaborate. The diagnosis is sound and well-motivated, and the proposal is quite compelling.

**Confidence Of Review:**

3

**Detailed Feedback Points:**

- The paper identifies that the obstacle to clinical AI in brain MRI is the data infrastructure rather than the models, and the five problems it lists (heterogeneity, governance, irreproducible preprocessing, weak metadata/\provenance, and demographic bias) are real and familiar to anyone who has worked with these types of data. It also works in the paper's favor that the proposed prototype is deliberately small and achievable.
- The paper is almost entirely diagnosis with no demonstration. The proof-of-concept is described but not built or evaluated, so there are no results indicating whether the proposed harmonization actually works. More preliminary data would be very helpful for intepreting this proposal.
- The central open question is how to verify that harmonized metadata preserves clinical meaning. It would be valuable to demonstrate that mapped fields such as severity scores and phenotype definitions agree with the source instruments, ideally with a clinician or domain expert in the loop. It would also strengthen the core argument to test whether harmonization improves a downstream classifier, which would connect the proposed infrastructure back to the diagnostic performance the paper is ultimately concerned with. In my personal experience, information is nearly always lost in the translation from source metadata -> harmonized data. How would the authors propose addressing this?

**Relevance For Biodms:**

3